# Disjoint Mapping Network for Cross-modal Matching of Voices and Faces

**Yandong Wen**[†]**, Mahmoud Al Ismail**[†]**, Weiyang Liu**[§]**, Bhiksha Raj**[†]**, Rita Singh**[†]
[†]Carnegie Mellon University    [§]Georgia Institute of Technology
yandongw@andrew.cmu.edu, mahmoudi@andrew.cmu.edu, wyliu@gatech.edu

## Abstract

We propose a novel framework, called Disjoint Mapping Network (DIMNet), for cross-modal biometric matching, in particular of voices and faces. Different from the existing methods, DIMNet does not explicitly learn the joint relationship between the modalities. Instead, DIMNet learns a shared representation for different modalities by mapping them individually to their common covariates. These shared representations can then be used to find the correspondences between the modalities. We show empirically that DIMNet is able to achieve better performance than the current state-of-the-art methods, with the additional benefits of being conceptually simpler and less data-intensive. The code is made available at https://github.com/ydwen/DIMNet.

## 1 Introduction

A person's face is predictive of their voice. Biologically, the genetic, physical and environmental influences that affect the face also affect the voice. Humans have been shown to be able to associate voices of unknown individuals to pictures of their faces (Kamachi et al., 2003). Humans also show improved ability to memorize and recall voices when previously exposed to pictures of the speaker's face, but not imposter faces (McAllister et al., 1993; Schweinberger et al., 2007; 2011). Cognitively, studies indicate that neuro-cognitive pathways for voices and faces share common structure (Ellis, 1989), possibly following parallel pathways within a common recognition framework (Belin et al., 2004; 2011). The above studies lend credence to the hypothesis that it may be possible to find associations between voices and faces algorithmically as well. With this in perspective, this paper focuses on the task of devising computational mechanisms for cross-modal matching of voice recordings and images of the speakers' faces.

The specific problem we look at is the one wherein we have an existing database of samples of people's voices and images of their faces, and we aim to automatically and accurately determine which voices match to which faces. This problem has seen significant research interest, in particular since the recent introduction of the `VoxCeleb` corpus (Nagrani et al., 2017), which comprises collections of video and audio recordings of a large number of celebrities. The existing approaches (Nagrani et al., 2018b;a; Kim et al., 2018) have generally attempted to directly relate subjects' voice recordings and their face images, in order to find the correspondences between the two. Nagrani et al. (2018b) formulates the mapping as a binary selection task: given a voice recording, one must successfully select the speaker's face from a pair of face images (or the reverse – given a face image, one must correctly select the subject's voice from a pair of voice recordings). They model the mapping as a neural network that is trained through joint presentation of voices and faces to determine if they belong to the same person. In Kim et al. (2018); Nagrani et al. (2018a), the authors attempt to learn common embeddings (*i.e.*, vector representations) for voices and faces that can be compared to one another to identify associations. The networks that compute the embeddings are also trained through joint presentation of voices and faces, to maximize the similarity of embeddings derived from them if they belong to the same speaker. In all cases, the voice and face are implicitly assumed to directly inform about one another.

In reality, though, it is unclear how much these models capture the direct influence of the voice and face on one another, and how much is explained through implicit capture of higher-level variables such as gender, age, ethnicity etc., which individually predict the two. These higher-level variables,

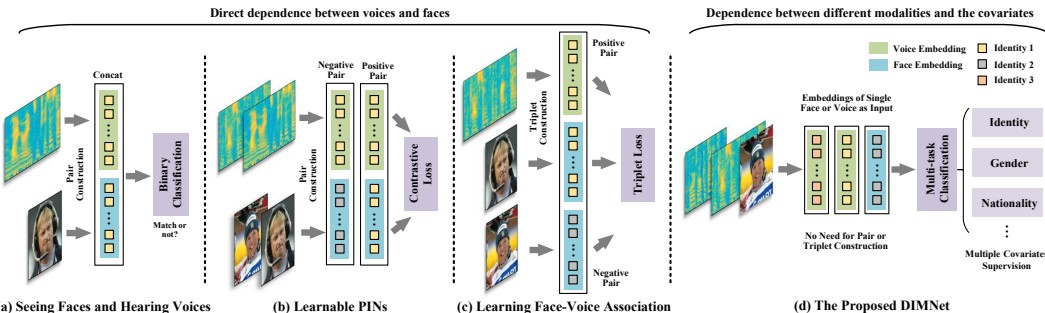

Figure 1: Overview of the proposed DIMNet and its comparison to the existing approaches. (a) Seeing faces and hearing voices from Nagrani et al. (2018b). (b) Learnable PINs from Nagrani et al. (2018a). (c) Learning face-voice association from Kim et al. (2018). (d) Our proposed DIMNets. DIMNets present a joint voice-face embedding framework via multi-task classification and require no pair construction (*i.e.*, both voices and faces can be input sequentially without forming pairs).

which we will refer to as *covariates*[1] can, in fact, explain much of our ability to match voices to faces (and vice versa) under the previously mentioned "select-from-a-pair" test (where a voice must be used to distinguish the speaker's face from a randomly-chosen imposter). For instance, simply matching the gender of the voice and the face can result in an apparent accuracy of match of up to 75% in a gender-balanced testing setting. Even in a seemingly less constrained "verification" test, where one must only verify if a given voice matches a given face, matching them based on gender alone can result in an equal error rate of 33% (Appendix B). Even matching the voice and the face by age (*e.g.* matching older-looking faces to older-sounding voices) could result in match accuracy that's significantly better than random.

Previous studies (Nagrani et al., 2018b; Kim et al., 2018) attempt to disambiguate the effect of multiple covariates through stratified tests that separate the data by covariate value. The results show that at least some of the learned associations are explained by the covariate, indicating that their learning approaches do utilize the covariate information, albeit only implicitly.

In this paper, we propose a novel framework to learn mappings between voices and faces that do not consider any direct dependence between the two, but instead explicitly exploit their individual dependence on the covariates. We define covariate as the identity-sensitive factors that can simultaneously affect voice and face, e.g. nationality, gender, identity (ID), etc. We do not require the *value* these factors take to be the same between the training and test set, since what we are learning is the nature of the covariation with the variable in general, not merely the covariation with the specific values the variable takes in the training set. In contrast to existing methods where supervision is provided through the correspondence of voices and faces, our learning framework, Disjoint Mapping Network (DIMNet), obtains supervision from common covariates, applied *separately* to voices and faces, to learn common embeddings for the two. The comparison between the existing approaches and DIMNets are illustrated in Fig. 1.

DIMNet comprises individual feature learning modules which learn identically-dimensioned features for data from each modality, and a unified input-modality-agnostic classifier that attempts to predict covariates from the learned feature. Data from each modality are presented separately during learning; however the unified classifier forces the feature representations learned from the individual modalities to be comparable. Once trained, the classifier can be removed and the learned feature representations are used to compare data across modalities.

The proposed approach greatly simplifies the learning process and, by considering the modalities individually rather than as coupled pairs, makes much more effective use of the data. Moreover, if multiple covariates are known, they can be simultaneously used for the training through multi-task learning in our framework (see Fig. 2).

Compared to current methods (Nagrani et al., 2018b;a; Kim et al., 2018), DIMNets achieve consistently better performance, indicating that direct supervision through covariates is more effective in

---

[1]To be clear, these are co*variates*, factors that vary jointly with voice and face, possibly due to some other common causative factors such as genetics, environment, *etc*. They are usually not claimed to be causative factors themselves.

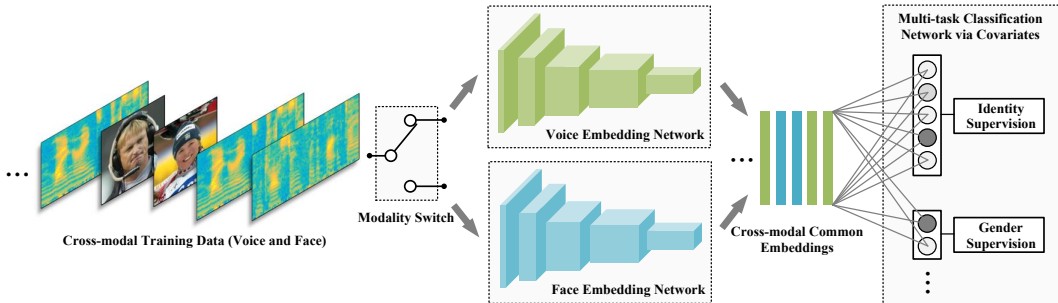

Figure 2: Our DIMNet framework. The input training data can be either voice or face, and there is no need for voices and faces to form pairs. Modality switch is to control which embedding network (voice or face) to process the data. While the embeddings are obtained, a multi-task classification network is applied to supervise the learning.

these settings. We find that of all the covariates, ID provides the strongest supervision. The results obtained from supervision through other covariates also match what may be expected.

Our contributions are summarized as follows:

- We propose DIMNets, a framework that formulates the problem of cross-modal matching of voices and faces as learning common embeddings for the two through individual supervision from one or more covariates, in contrast to current approaches that attempt to map voices to faces directly. An overview of our framework is given in Fig. 2.
- In this framework, we can make full use of multiple kinds of label information (provided by covariates) with a multi-task objective function.
- We achieve the state-of-the-art results on multiple tasks. We are also able to isolate and analyze the effect of the individual covariate on the performance.

Moreover, we note that the proposed framework is applicable in any setting where matching of different types of data which have common covariates is required.

## 2 THE PROPOSED FRAMEWORK

Our goal is to learn common vector representations for both voices and faces, that permit them to be compared to one another. In the following sections we first describe how we learn them from their relationship to common *covariates*. Subsequently, we describe how we will use them for comparison of voices to faces.

### 2.1 LEVERAGING COVARIATES TO LEARN EMBEDDINGS

The relationship between voices and faces is largely predicted by *covariates* – factors that individually relate to both the voice and the face. To cite a trivial example, a person's gender relates their voice to their face: male subjects will have male voices and faces, while female subjects will have female voices and faces. More generally, many covariates may be found that relate to both voice and face (Lippert et al., 2017).

Our model attempts to find common representations for both face images and voice recordings by leveraging their relationship to these covariates (rather than to each other). We will do so by attempting to predict covariates from voice and face data in a common embedding space, such that the derived embeddings from the two types of data can be compared to one another.

Let $\mathcal{V}$ represent a set of voice recordings, and $\mathcal{F}$ represent a set of face images. Let $\mathcal{C}$ be the set of covariates we consider. For the purpose of this paper, we assume that all covariates are discrete valued (although this is not necessary). Every voice recording in $\mathcal{V}$ and every face in $\mathcal{F}$ can be related to each of the covariates in $\mathcal{C}$. For every covariate $C \in \mathcal{C}$ we represent the value of that covariate for any voice recording $v$ as $C(v)$, and similarly the value of the covariate for any face $f$ as $C(f)$. For example, $C$ could be ID, gender, or nationality. When $C$ is ID, $C(v)$ and $C(f)$ are the ID of voice $v$ and face $f$, respectively.

Let $F_v(v; \theta_v) : v \mapsto \mathbb{R}^d$ be a *voice embedding* function with parameters $\theta_v$ that maps any voice recording $v$ into a $d$-dimensional vector. Similarly, let $F_f(f; \theta_f)$ be a *face embedding* function that

maps any face $f$ into a $d$-dimensional vector. We aim to learn $\theta_v$ and $\theta_f$ such that the embeddings of the voice and face for any person are comparable.

For each covariate $C \in \mathcal{C}$ we define a classifier $H_C(x; \phi_C)$ with parameter $\phi_C$, which assigns any input $x \in \mathbb{R}^d$ to one of the values taken by $C$. The classifier $H_C(\cdot)$ is agnostic to which modality its input $x$ was derived from; thus, given an input voice $v$, it operates on features $F_v(v; \theta_v)$ derived from the voice, whereas given a face $f$, it operates on $F_f(f; \theta_f)$.

For each $v$ (or $f$) and each covariate $C$, we define a loss $L(H_C(F_v(v; \theta_v); \phi_C), C(v))$ between the covariate predicted by $H_C(.)$ and the true value of the covariate for $v$, $C(v)$. We can now define a *total loss* $\mathcal{L}$ over the set of all voices $\mathcal{V}$ and the set of all faces $\mathcal{F}$, over all covariates as

$$\mathcal{L}(\theta_v, \theta_f, \{\phi_C\}) = \sum_{C \in \mathcal{C}} \lambda_C \Bigg( \sum_{v \in \mathcal{V}} L(H_C(F_v(v; \theta_v); \phi_C), C(v)) $$
$$+ \sum_{f \in \mathcal{F}} L(H_C(F_f(f; \theta_f); \phi_C), C(f)) \Bigg) \tag{1}$$

In order to learn the parameters of the embedding functions, $\theta_f$ and $\theta_v$, we perform the following optimization.

$$\theta_v^*, \theta_f^* = \arg \min_{\theta_v, \theta_f} \min_{\{\phi_C\}} \mathcal{L}(\theta_v, \theta_f, \{\phi_C\}) \tag{2}$$

## 2.2 DISJOINT MAPPING NETWORKS

In DIMNet, we instantiate $F_v(v; \theta_v)$, $F_f(f; \theta_f)$ and $H_C(x; \phi_C)$ as neural networks. Fig. 2 shows the network architecture we use to train our embeddings. It comprises three components. The first, labelled *Voice Network* in the figure, represents $F_v(v; \theta_v)$ and is a neural network that extracts $d$-dimensional embeddings of the voice recordings. The second, labelled *Face Network* in the figure, represents $F_f(f; \theta_f)$ and is a network that extracts $d$-dimensional embeddings of face recordings. The third component, labelled *Classification Networks* in the figure, is a bank of one or more classification networks, one per covariate considered. Each of the classification networks operates on the $d$-dimensional features output by the embedding networks to classify one covariate, *e.g.* gender.

The training data comprise voice recordings and face images. Voice recordings are sent to the voice-embedding network, while face images are sent to the face-embedding network. This switching operation is illustrated by the switch at the input in Fig.2. In either case, the output of the embedding network is sent to the covariate classifiers.

As can be seen, at any time the system either operates on a voice, or on a face, *i.e.* the operations on voices and faces are disjoint. During the learning phase too, the updates of the two networks are disjoint – loss gradients computed when the input is voice only update the voice network, while loss gradients derived from face inputs update the face network, while both contribute to updates of the classification networks.

In our implementation, specifically, $F_v(\cdot)$ is a convolutional neural network that operates on Mel-Spectrographic representations of the speech signal. The output of the final layer is pooled over time to obtain a final $d$-dimensional representation. $F_f(\cdot)$ is also a convolutional network with a pooled output at the final layer that produces a $d$-dimensional representation of input images. The classifiers $H_C(\cdot)$ are all simple multi-class logistic-regression classifiers comprising a single softmax layer.

Finally, in keeping with the standard paradigms for training neural network systems, we use the cross-entropy loss to optimize the networks. Also, instead of the optimization in Eq. 2, the actual optimization performed is the one below. The difference is inconsequential.

$$\theta_v^*, \theta_f^*, \{\phi_C^*\} = \operatorname*{argmin}_{\theta_v, \theta_f, \{\phi_C\}} \mathcal{L}(\theta_v, \theta_f, \{\phi_C\}) \tag{3}$$

## 2.3 TRAINING THE DIMNET

All parameters of the network are trained through backpropagation, using stochastic gradient descent. During training, we construct the minibatches with a mixture of speech segments and face images, as the network learns more robust cross-modal features with mixed inputs. Taking voice as an example, we compute the voice embeddings using $F_v(v; \theta_v)$, and obtain the losses using classifiers $H_C(\cdot)$ for all the covariates. We back-propagate the loss gradient to update the voice network as

well as the covariate classifiers. The same procedure is also applied to face data: the backpropagated loss gradients are used to update the face network and the covariate classifiers. Thus, the embedding functions are learned using the data from their modalities individually, while the classifiers are learned using data from all modalities.

## 2.4 USING THE EMBEDDINGS

Once trained, the embedding networks $F_v(v; \theta_v)$ and $F_f(f; \theta_f)$ can be used to extract embeddings from any voice recording or face image. Given a voice recording $v$ and a face image $f$, we can now compute a similarity between the two through the cosine similarity $S(v, f) = \frac{F_v^\top F_f}{\|F_v\|_2 \|F_f\|_2}$. We can employ this similarity to evaluate the match of any face image to any voice recording. This enables us, for instance, to attempt to rank a collection of faces $f_1, \cdots, f_K$ in order of estimated match to a given voice recording $v$, according to $S(v, f_i)$, or conversely, to rank a collection of voices $v_1, \cdots, v_K$ according to their match to a face $f$, on order of decreasing $S(v_i, f)$.

## 3 EXPERIMENTS

We ran experiments on matching voices to faces, to evaluate the embeddings derived by DIMNets. The details of the experiments are given below and Appendix A.

**Datasets.** Our experiments were conducted on the `Voxceleb` (Nagrani et al., 2017) and `VGGFace` (Parkhi et al., 2015) datasets, which are specified in appendix A.1. We use the intersection of the two datasets, *i.e.*subjects who figure in both corpora, for our final corpus, which thus includes 1,225 IDs with 667 males and 558 females from 36 nationalities. The data are split into train/validation/test sets, following the settings in Nagrani et al. (2018b). Details can be found in Appendix A.1. We use ID, gender and nationality as our covariates, all of which are provided by the datasets. Separated data preprocessing pipelines are employed to audio segments and face images (see Appendix A.2).

**Training.** The detailed network configurations are elaborated in appendix A.3. Note that the classification networks are single-layer softmax units with as many outputs as the number of unique values the class can take (2 for gender, 32 for nationalities, and 924 for IDs in our case). The networks are trained to minimize the cross entropy loss, following the typical settings of stochastic gradient descent (SGD) in appendix A.3

**Testing.** We use the following protocols for evaluation:

- *1:2 Matching.* Here, we are given a probe input from one modality (voice or face), and a gallery of two inputs from the other modality (face or voice), including one that belongs to the same subject as the probe, and another of an "imposter" that does not match the probe. The task is to identify which entry in the gallery matches the probe. We report performance in terms of matching accuracy – namely what fraction of the time we correctly identify the right instance in the gallery.

  To minimize the influence of random selection, we construct as many testing instances as possible through exhaustive enumeration all positive matched pairs (of voice and face). To each pair, we include a randomly drawn imposter in the gallery. We thus have a total of 4,678,897 trials in the validation set, and 6,780,750 trials in the test set.

- *1:N Matching.* This is the same as the 1:2 matching, except that the gallery now includes $N - 1$ imposters. Thus, we must now identify which of the $N$ entries in the gallery matches the probe. Here too results are reported in terms of matching accuracy. We use the same validation and test sets as the 1:2 case, by augmenting each trial with $N - 2$ additional imposters. So the number of trials in validation and test sets is the same as earlier.

- *Verification.* We are given two inputs, one a face, and another a voice. The task is to determine if they are matched, *i.e.*both belong to the same subject. In this problem setting the similarity between the two is compared to a threshold to decide a match. The threshold can be adjusted to trade off *false rejections* ($F_R$), *i.e.*wrongly rejecting true matches, with *false alarms* ($F_A$), *i.e.*wrongly accepting mismatches. We report results in terms of *equal error rate*, *i.e.*when $F_R = F_A$. We construct our validation and test sets from those used for the 1:2 matching tests, by separating each trial into two, one comprising a matched pair, and the other a mismatched pair. Thus, our validation and test sets are exactly twice as large as those for the 1:2 test.

- *Retrieval.* The gallery comprises a large number of instances, one or *more* of which might match the probe. The task is to order the gallery such that the entries in the gallery that match the probe lie at the top of the ordering. Here, we report performance in terms of *Mean Average Precision* (MAP) (Manning et al., 2008). Here we use the *entire* collection of 58,420 test faces as the gallery for each of our 21,799 test voices, when retrieving faces from voices. For the reverse (retrieving voices from faces), the numbers are reversed.

Each result is obtained by averaging the performances of 5 models, which are individually trained.

**Covariates in Training and Testing.** We use the three covariates provided in the dataset, namely identity (I), gender (G), and nationality (N) for our experiments. The treatment of covariates differs for training and test.

- *Training.* For training, supervision may be provided by any set of (one two or three) covariates. We consider all combinations of covariates, I, G, N, (I,G), (I,N), (G,N) and (I,G,N). Increasing the number of covariates effectively increases the supervision provided to training. All chosen covariates were assigned a weight of 1.0.
- *Testing.* As explained in Appendix B, simply recognizing a covariate such as gender can result in seemingly significant matching performance. For instance, just recognizing the subjects' gender from their voice and images can result in a 33% EER for verification, and 25% error in matching for the 1 : 2 tests. In order to isolate the effect of covariates on performance hence we also *stratify* our test data by them. Thus we construct 4 testing groups based on the covariates, including the unstratified (U) group, stratified by gender (G), stratified by nationality (N), and stratified by gender and nationality (G, N). In each group the test set itself is separated into multiple strata, such that for all instances within any stratum the covariate values are the same.

## 3.1 CROSS-MODAL MATCHING

In this section we report results on the 1:2 and 1:$N$ matching tests. In order to ensure that the embedding networks do indeed leverage on accurate modelling of covariates, we first evaluate the classification accuracy of the classification networks for the covariates themselves. Table 1 shows the results.

| method | gender classification | | nationality classification | |
|---|---|---|---|---|
| | voice | face | voice | face |
| DIMNet-I | - | - | - | - |
| DIMNet-G | 97.48 | 99.22 | - | - |
| DIMNet-N | - | - | **74.86** | 60.13 |
| DIMNet-IG | **97.70** | **99.42** | - | - |
| DIMNet-IN | - | - | 74.17 | 60.27 |
| DIMNet-GN | 97.59 | 99.06 | 74.62 | **60.50** |
| DIMNet-IGN | 97.69 | 99.15 | 74.37 | 59.88 |

Table 1: Acc. (%) of covariate prediction.

The rows of the table show the covariates used to supervise the learning. Thus, for instance, the row labelled "DIMNet-I" shows results obtained when the networks have been trained using ID alone as covariate, the row labelled "DIMNet-G" shows results when supervision is provided by gender, "DIMNet-IG" has been trained using ID and gender, etc.

The columns of the table show the specific covariate being evaluated. Since the identities of subjects in the training and test set do not overlap, we are unable to evaluate the accuracy of ID classification. Note that we can only test the accuracy of the classification network for a covariate if it has been used in the training. Thus, classification accuracy for gender can be evaluated for DIMNet-G, DIMNet-GN and DIMNet-IGN, while that for nationality can be evaluated for DIMNet-N, DIMNet-GN and DIMNet-IGN.

The results in Table 1 show that gender is learned very well, and in all cases gender recognition accuracy is quite high. Nationality, on the other hand, is not a well-learned classifier, presumably because the distribution of nationalities in the data set is highly skewed (Nagrani et al., 2018b), with nearly 65% of all subjects belonging to the USA. It is to be expected therefore that nationality as a covariate will not provide sufficient supervision to learn good embeddings.

**1:2 matching.** Table 2 shows the results for the 1:2 matching tests. In the table, the row labelled "SVHF-Net" gives results obtained with the model of Nagrani et al. (2018b).

The columns are segregated into two groups, one labelled "voice → face" and the other labelled "face → voice". In the former, the probe is a voice recording, while the gallery comprises faces. In the later the modalities are reversed. Within each group the columns represent the stratification of the test set. "U" represents test sets that are not stratified, and include the various covariates in the same proportion that they occur in the overall test set. The columns labelled "G" and "N" have been stratified by gender and nationality, respectively, while the column "G, N" represents data that

| method | voice → face (ACC %) | | | | face → voice (ACC %) | | | |
|---|---|---|---|---|---|---|---|---|
| | U | G | N | G, N | U | G | N | G, N |
| SVHF-Net | 81.00 | 63.90 | - | - | 79.50 | 63.40 | - | - |
| DIMNet-I | 83.45±0.42 | 70.91±0.56 | 81.97±0.51 | 69.89±0.78 | 83.52±0.45 | 71.78±0.55 | 82.41±0.48 | **70.90**±0.81 |
| DIMNet-G | 72.90±0.55 | 50.32±0.70 | 71.92±0.51 | 50.21±0.65 | 72.47±0.54 | 50.48±0.71 | 72.15±0.54 | 50.61±0.68 |
| DIMNet-N | 57.53±0.45 | 55.33±0.67 | 53.04±0.43 | 51.96±0.59 | 56.20±0.43 | 54.34±0.61 | 53.90±0.44 | 51.97±0.57 |
| DIMNet-IG | **84.12**±0.44 | **71.32**±0.60 | **82.65**±0.57 | **70.39**±0.80 | **84.03**±0.39 | **71.65**±0.60 | **82.96**±0.49 | 70.78±0.47 |
| DIMNet-IN | 82.95±0.40 | 70.04±0.67 | 81.04±0.55 | 68.59±0.76 | 82.86±0.35 | 70.91±0.59 | 81.91±0.52 | 70.22±0.77 |
| DIMNet-GN | 75.92±0.42 | 56.66±0.55 | 72.94±0.48 | 53.48±0.73 | 73.78±0.69 | 54.90±0.54 | 72.63±0.48 | 53.45±0.85 |
| DIMNet-IGN | 83.73±0.53 | 70.76±0.34 | 81.75±0.48 | 69.17±0.71 | 83.63±0.66 | 71.42±0.49 | 82.50±0.43 | 70.46±0.62 |

Table 2: Performance comparison of 1:2 matching for models trained using different sets of covariates.

have been stratified by both gender and nationality. In the stratified tests, we have ensured that all data within a test instance have the same value for the chosen covariate. Thus, for instance, in a test instance for voice → face in the "G" column, the voice and both faces belong to the same gender. This does not reduce the overall number of test instances, since it only requires ensuring that the gender of the imposter matches that of the probe instance.

We make several observations. First, DIMNet-I performs better than SVHF-Net, improving the accuracies by 2.45%-4.02% for the U group, and 7.01%-8.38% for the G group. It shows that mapping voices and faces to their common covariates is an effective strategy to learn representations for cross-modal matching.

Second, DIMNet-I produces significantly better embeddings that DIMNet-G and DIMNet-N, highlighting the rather unsurprising fact that ID provides the more useful information than the other two covariates. In particular, DIMNet-G respectively achieves 72.90% and 72.47% for voice to face and face to voice matching using only gender as a covariate. This verifies our hypothesis that we can achieve almost 75% matching accuracy by only using the gender. These numbers also agree with the performance expected from the numbers in Table 1 and the analysis in Appendix B. As expected, nationality as a covariate does not provide as good supervision as gender. DIMNet-IG is marginally better than DIMNet-I, indicating that gender supervision provides additional support over ID alone.

Third, we note that while DIMNet-I is able to achieve good performance on the dataset stratified by gender, DIMNet-G only achieves random performance. The performance achieved by DIMNet-G on the U dataset is hence completely explained by gender matching. Once again, the numbers match our expectations (Appendix B).

**1:N matching.** We also experiment for $N > 2$. Unlike SVHF-Net (Nagrani et al., 2018b) that needs to train different models for different $N$ in this setting, we use the same model for different $N$. The results in Fig. 3 shows accuracy as a function of $N$ for various models.

All the results in Fig. 3 are consistent with Table 2. As expected, the performance of all methods degrades with increasing $N$. In general, DIMNets that use ID as supervision outperform SVHF-Net by a considerable margin, showing that DIMNets are able to make best use of the ID information. We obtain the best results when both ID and gender are used as supervision covariates. However, The results obtained using only gender information as covariate is much worse, which is also consistent with our analysis in Appendix B.

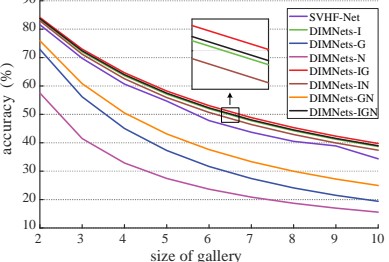

Figure 3: Performance of 1:$N$ matching

## 3.2 Cross-modal Verification

For verification, we need to determine whether an audio segment and a face image are from the same ID or not. We report the equal error rate (EER) for verification in Table 3.

In general, DIMNets that use ID as a covariate achieve an EER of about 25%, which is considerably lower than the 33% expected if the verification were based on gender matching alone. The

| method | verification (EER %) | | | |
|---|---|---|---|---|
| | U | G | N | G, N |
| DIMNet-I | 24.95±0.20 | 34.95±0.45 | 25.92±0.68 | 35.74±0.87 |
| DIMNet-G | 34.86±0.11 | 49.69±0.24 | 35.13±0.36 | 49.67±0.51 |
| DIMNet-N | 45.89±0.39 | 46.97±0.55 | 47.89±0.82 | 48.87±1.14 |
| DIMNet-IG | **24.56**±0.23 | **34.84**±0.41 | **25.54**±0.65 | **35.73**±0.79 |
| DIMNet-IN | 25.54±0.18 | 36.22±0.40 | 27.25±0.72 | 37.39±0.79 |
| DIMNet-GN | 33.28±0.52 | 46.65±0.16 | 34.77±0.26 | 48.08±0.52 |
| DIMNet-IGN | 25.00±0.19 | 35.76±0.36 | 26.80±0.69 | 37.30±0.74 |

Table 3: Verification results.

results in Table 3 show that using both gender and ID information as covariates can further improve the performance over using ID alone, well validating the superiority of our multi-task learning framework. Using proper combination of covariates is crucial to the performance. ID is arguably

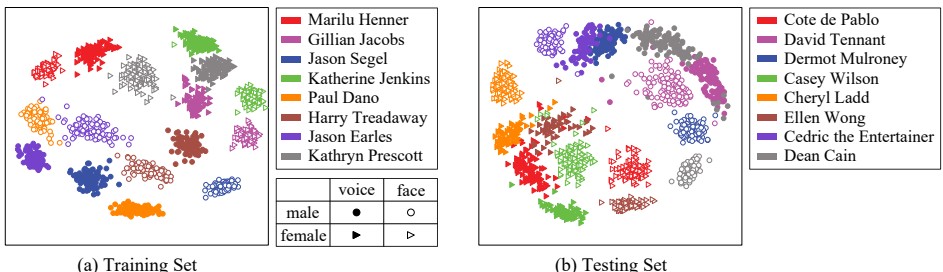

(a) Training Set      (b) Testing Set

Figure 4: Visualization of voice and face embeddings using multi-dimensional scaling Wickelmaier (2003) . The left panel shows subjects from the training set, while the right panel is from the test set.

the most effective covariate supervision. More interestingly, nationality is seen to be an ineffective covariate, while gender alone as a covariate produces results that well matches our expectation.

## 3.3 CROSS-MODAL RETRIEVAL

We also perform retrieval experiments using voice or face as query. Table 4 lists the mean average precision (mAP) of the retrieval for various models.

The columns in the table represent the covariate being retrieved. Thus, for example, in the "ID" column, the objective is to retrieve

| method | voice → face (mAP %) | | | face → voice (mAP %) | | |
|---|---|---|---|---|---|---|
| | ID | gender | nationality | ID | gender | nationality |
| Random | 0.58 | 52.61 | 40.70 | 0.55 | 52.60 | 40.69 |
| DIMNet-I | 4.25±0.11 | 89.57±0.35 | 43.26±0.16 | 4.17±0.10 | 88.50±0.37 | 43.68±0.20 |
| DIMNet-G | 1.07±0.07 | **97.84**±0.58 | 41.56±0.23 | 1.15±0.12 | **97.15**±0.62 | 41.97±0.20 |
| DIMNet-N | 1.24±0.13 | 56.99±0.32 | **45.69**±0.65 | 1.03±0.10 | 56.90±0.34 | **49.30**±0.57 |
| DIMNet-IG | **4.42**±0.12 | 93.10±0.45 | 43.22±0.14 | **4.23**±0.09 | 92.16±0.42 | 43.86±0.17 |
| DIMNet-IN | 3.94±0.11 | 89.72±0.39 | 43.95±0.69 | 3.99±0.14 | 88.39±0.39 | 45.93±0.66 |
| DIMNet-GN | 1.89±0.11 | 95.89±0.44 | 45.20±0.63 | 1.64±0.11 | 93.95±0.43 | 48.39±0.54 |
| DIMNet-IGN | 4.07±0.09 | 92.30±0.57 | 44.10±0.62 | 4.05±0.09 | 91.31±0.61 | 45.82±0.59 |

Table 4: Retrieval performance (mAP).

gallery items with the same ID as the query, whereas in the "gender" column the objective is to retrieve the same gender.

We note that ID-based DIMNets produce the best features for retrieval, with the best performance obtained with DIMNet-IG. Also, as may be expected, the covariates used in training result in the best retrieval of that covariate. Thus, DIMNet-G achieves an mAP of nearly 98% on gender, though on retrieval of ID it is very poor. As in other experiments, nationality remains a poor covariate in general. Compared to gender (2 classes) and nationality (unbalanced 28 classes), retrieving ID is a challenging problem given the large amount of identities (182 classes). The significant and consistent improvements over chance-level results show that the DIMNet models do learn some useful associations between voices and faces.

## 3.4 COMPARISONS TO THE CURRENT STATE-OF-THE-ART

| | Seen-Heard | | | | | Unseen-Unheard | | | | |
|---|---|---|---|---|---|---|---|---|---|---|
| | U | G | N | A | G, N, A | U | G | N | A | G, N, A |
| Nagrani et al. (2018a) | 87.0 | 74.2 | 85.9 | 86.6 | 74.0 | 78.5 | 61.1 | 77.2 | 74.9 | 58.8 |
| DIMNet-I | **95.1**±0.23 | **90.8**±0.25 | **93.4**±0.15 | **95.2**±0.11 | **88.9**±0.21 | 82.5±0.12 | 71.0±0.33 | 81.1±0.10 | 77.7±0.14 | **62.8**±0.36 |
| DIMNet-IG | 94.7±0.23 | 89.8±0.22 | 93.2±0.13 | 94.8±0.12 | 87.8±0.18 | **83.2**±0.11 | **71.2**±0.37 | **81.9**±0.18 | **78.0**±0.13 | **62.8**±0.39 |

Table 5: AUCs (%) of DIMNets under different testing groups.

We compare DIMNet with the state of the art (Nagrani et al., 2018a). The results are reported in Table 5. Note that it is fair comparison because the DIMNet models in this section are trained with and evaluated on the same released datasets in Nagrani et al. (2018a). Detailed statistics and splits of the dataset can be found in Appendix A.1. There are two evaluation protocols, including Seen-Heard and Unseen-Unheard scenarios. The identities of the training and testing set have overlaps in Seen-Heard scenario (closed-set), while they are fully disjoint in Unseen-Unheard scenario (open-set). For each scenario, there are 5 testing groups based on the covariates, including the unstratified group (U), group, stratified by gender (G), stratified by nationality (N), stratified by age (A), and stratified by (G, N, A). We compute the area under the curve (AUC) for different testing groups.

It is clear that DIMNets produce better embeddings than Nagrani et al. (2018a) for pair-wise verification on both seen-heard and unseen-unheard scenarios. Specifically, DIMNets achieve 8%-15% absolute and 3%-10% absolute improvements on seen-heard and unseen-unheard test set, respectively.

Compared to DIMNet-IG, DIMNet-I performs better on the seen-heard test set while DIMNet-IG is better on the unseen-unheard test set. It implies that introducing useful covariates improves the generalization capability of DIMNet.

## 4 DISCUSSIONS AND CONCLUDING REMARKS

We have proposed that it is possible to learn common embeddings for multi-modal inputs, particularly voices and faces, by mapping them individually to common covariates. In particular, the proposed DIMNet architecture is able to extract embeddings for both modalities that achieves consistently better performance than the methods that directly map faces to voices.

The approach also provides us the ability to tease out the influence of each of the covariates of voice and face data, in determining their relation. The results show that the strongest covariate, not unexpectedly, is ID. The results also indicate that prior results by other researchers who have attempted to directly match voices to faces may perhaps not be learning any direct relation between the two, but implicitly learning about the common covariates, such as ID, gender, etc.

Our experiments also show that although we have achieved possibly the best reported performance on this task, thus far, the performance is not anywhere close to prime-time. In the $1 : N$ matching task, performance degrades rapidly with increasing $N$, indicating a rather poor degree of true match.

To better understand the problem, we have visualized the learned embeddings from DIMNet-I in Fig. 4 to provide more insights. The visualization method we used is multi-dimensional scaling (MDS) (Wickelmaier, 2003), rather than the currently more popular t-SNE (van der Maaten & Hinton, 2008). This is because MDS tends to preserve distances and global structure, while t-SNE attempts to retain statistical properties and highlights clusters, but does not preserve distances.

From Fig. 4, we immediately notice that the voice and face data for a subject are only weakly proximate. While voice and face embeddings for a speaker are generally relatively close to each other, they are often closer to other subjects. Interestingly, the genders separate (even though gender has not been used as a covariate for this particular network), showing that at least *some* of the natural structure of the data is learned. Fig. 4 shows embeddings obtained from both training and test data. We can observe similar behaviors in both, showing that the the general characteristics observed are not just the outcome of overfitting to training data. The visualization in Fig. 4 also shows that there is still significant room for improvement. For example, it may be possible to force compactness of the distributions of voice and face embeddings through modified loss functions such as the center loss (Wen et al., 2016) or angular softmax loss (Liu et al., 2016; 2017a;b), or through an appropriately designed loss function that is specific to this task.

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

## APPENDIX A    EXPERIMENTAL DETAILS

### A.1    DATASET

The `Voxceleb` dataset consists of 153,516 audio segments from 1,251 speakers. Each audio segment is taken from an online video clip with an average duration of 8.2 seconds. For the face dataset, we used a manually filtered version of `VGGFace`. After face detection, there remain 759,643 images from 2,554 subjects. The data are split into train/validation/test sets, following the settings in Nagrani et al. (2018b). Details are shown in Table 6

| # of samples | train | validation | test | total |
|---|---|---|---|---|
| speech segments | 112,697 | 14,160 | 21,799 | 148,656 |
| face images | 313,593 | 36,716 | 58,420 | 408,729 |
| IDs | 924 | 112 | 189 | 1,225 |
| genders | 2 | 2 | 2 | 2 |
| nationalities | 32 | 11 | 18 | 36 |
| testing instances | - | 4,678,897 | 6,780,750 | 11,459,647 |

Table 6: Statistics for the data appearing in `VoxCeleb` and `VGGFace`.

The visual data used in Section 3.4 is densely extracted from the video in `VoxCeleb` dataset at 25/6 fps. It contains 100,000 segmented speaking face-tracks obtained by SyncNet (Chung & Zisserman, 2016), leading to 1,218,575 frames (images). For fair comparison, we follow the train/val/test split strategy from Nagrani et al. (2018a) in our experiments. The evaluations are performed based on the provided lists (Nagrani et al., 2018a), which specify the testing pairs of voices and faces.

### A.2    PREPROCESSING

We employ separated data preprocessing pipelines for audio segments and face images. For audio segments, we use an energy-based voice activity detector (Povey et al., 2011) to isolate speech-bearing regions of the recordings. Subsequently, 64-dimensional log mel-spectrograms are generated, using an analysis window of 25ms, with hop of 10ms between frames. We perform mean and variance normalization of each mel-frequency bin.

For training, we randomly crop out regions of varying lengths of 300 to 800 frames (so the size of the input spectrogram ranges from $300 \times 64$ to $800 \times 64$ for each mini-batch, around 3 to 8 seconds). For the face data, facial landmarks in all images are detected using MTCNN (Zhang et al., 2016). The cropped RGB face images of size $128 \times 128 \times 3$ are obtained by similarity transformation. Each pixel in the RGB images is normalized by subtracting 127.5 and then dividing by 127.5. We perform data augmentation by horizontally flipping the images with $50\%$ probability in minibatches (effectively doubling the number of face images).

### A.3    TRAINING

The details of network architectures are shown in Table 7. For the voice network, we use 1D convolutional layers, where the convolution is performed along the axis that corresponds to time. The face network employs 2D convolutional layers. For both, the convolutional layers are followed by batch normalization (BN) (Ioffe & Szegedy, 2015) and rectified linear unit activations (ReLU) (Krizhevsky et al., 2012). The final face embedding is obtained by averaging the feature maps from the final layer, *i.e.*through *average pooling*. The final voice embedding is obtained by averaging the feature maps at the final convolutional layer along the time axis alone.

We follow the typical settings of SGD for optimization. Minibatch size is 256. The momentum and weight decay values are 0.9 and 0.001 respectively. To learn the networks from scratch, the learning rate is initialized at 0.1 and divided by 10 after 16K iterations and again after 24K iterations. The training is completed at 28K iterations.

| | layer | voice | face |
|---|---|---|---|
| embedding network | Conv | $(3, 256)_{/2,1}$
$\begin{bmatrix}(3, 256)_{/1,1}\\(3, 256)_{/1,1}\end{bmatrix}$ | $(3 \times 3, 64)_{/2,1}$
$\begin{bmatrix}(3 \times 3, 64)_{/1,1}\\(3 \times 3, 64)_{/1,1}\end{bmatrix}$ |
| | | $(3, 384)_{/2,1}$
$\begin{bmatrix}(3, 384)_{/1,1}\\(3, 384)_{/1,1}\end{bmatrix}$ | $(3 \times 3, 128)_{/2,1}$
$\begin{bmatrix}(3 \times 3, 128)_{/1,1}\\(3 \times 3, 128)_{/1,1}\end{bmatrix}$ |
| | | $(3, 576)_{/2,1}$
$\begin{bmatrix}(3, 576)_{/1,1}\\(3, 576)_{/1,1}\end{bmatrix}$ | $(3 \times 3, 256)_{/2,1}$
$\begin{bmatrix}(3 \times 3, 256)_{/1,1}\\(3 \times 3, 256)_{/1,1}\end{bmatrix}$ |
| | | $(3, 864)_{/2,1}$
$\begin{bmatrix}(3, 864)_{/1,1}\\(3, 864)_{/1,1}\end{bmatrix}$ | $(3 \times 3, 512)_{/2,1}$
$\begin{bmatrix}(3 \times 3, 512)_{/1,1}\\(3 \times 3, 512)_{/1,1}\end{bmatrix}$ |
| | | $(3, 64)_{/2,1}$ | $(3 \times 3, 64)_{/2,1}$ |
| | AvgPool | $t \times 1$ | $h \times w \times 1$ |
| classification network | FC | $64 \times 924, 64 \times 2, 64 \times 32$ | |

Table 7: The detailed CNNs architectures. The numbers within the parentheses represent the size and number of filters, while the subscripts represent the stride and padding. So, for example, $(3, 64)_{/2,1}$ denotes a 1D convolutional layer with 64 filters of size 3, where the stride and padding are 2 and 1 respectively, while $(3 \times 3, 64)_{/2,1}$ represents a 2-D convolutional layer of 64 $3 \times 3$ filters, with stride 2 and padding 1 in both directions. Note that 924, 2, and 32 are the number of unique values taken by the ID, gender, and nationality covariates, respectively.

### A.4 EXPERIMENTS ON THE EMBEDDING DIMENSION

To investigate the affect of embedding dimension to the performance, we train DIMNet-I models with various embedding dimensions of 32, 64, 128, 256, and 512. Table 8 shows the results on 1:2 matching experiment (voice → face).

| embedding dimension | 32 | 64 | 128 | 256 | 512 |
|---|---|---|---|---|---|
| DIMNet-I | 82.20 | 83.45 | 83.87 | 83.43 | 83.16 |

Table 8: The accuracies of DIMNet-I with different embedding dimensions on 1:2 matching experiments

It could be observed that the performance of cross-modal matching is very stable within a wide range of embedding dimension, showing that the accuracy is not sensitive to the embedding dimension.

## APPENDIX B  EXPECTED PERFORMANCE BASED ON GENDER MATCHING

In this appendix we discuss the performance to be expected in the matching and verification tests, when the matching is done based purely on gender.

We assume below that in any distribution of human-subject data, the division of subjects between male and female genders to be half and half.

It is to be noted that gender is merely an illustrative example here; the analysis can be extended to other covariates. For a more detailed analysis of covariates with more values and unbalanced distributions, please refer to Wen et al. (2018).

### B.1  ACCURACY OF 1:2 MATCHING BASED ON GENDER

We show that the equal-error-rate for 1:2 matching can be as high as 25%, through gender matching alone.

The problem is as follows: a probe input (voice or face), and a gallery consisting of two inputs (face or voice), one of which is from the same subject as the probe. We must identify which of the two is the true match.

### B.1.1 PERFECT GENDER IDENTIFICATION

Consider the situation where we are able to identify the gender of the subject of the data (face or voice) perfectly.

There are two possibilities: (a) both probe instances are the same gender, and (b) they are different genders. Each of the two possibilities occurs with a probability of 0.5

We employ the following simple strategy: If the two gallery instances are different genders, then we select the instance whose gender matches the probe. In this case, clearly, the probability of error is 0. If the two instances are the same gender, we select one of them randomly with a probability of 0.5. The probability of error here is 0.5.

Thus, the overall probability of error is

$$Prob(error) = 0.5 \times 0 + 0.5 \times 0.5 = 0.25.$$

### B.1.2 IMPERFECT GENDER IDENTIFICATION

Now let us consider the situation where gender identification itself is imperfect, and we have error rates $e_f$ and $e_v$ in identifying the gender of faces and voices, respectively. Assume the error rates are known. We will assume below that gallery entries are faces, and probe entries are voices. (The equations are trivially flipped to handle the converse case).

Since we are aware that we sometimes make mistakes in identifying gender, we modify our strategy as follows: when the two gallery items are found to have different genders, we select the entry with the same gender as the probe $P$ of the time (so that if the gender classification was correct, we would have a match error rate of $(1-P)$). When both gallery items are found to be the same gender, we choose randomly.

The actual error can now be computed as follows. The gallery items are both of the same gender in 0.5 of the trials, and of mismatched gender in the remaining 0.5 of the trials.

When both gallery items have the same gender, regardless of the strategy chosen, the probability of error is 0.5 (by symmetry).

When both gallery items are of mismatched gender, we have 8 combinations of correctness of gender-classification. Table 9 lists all eight, along with the probability of matching error (in the final column). Taking type 1 as an example, we have probability $(1 - e_v)(1 - e_f)^2$ that the gender

| type | probe | gallery1 | gallery2 | $Prob(error, type)$ |
|------|-------|----------|----------|---------------------|
| 1 | ✓ | ✓ | ✓ | $(1 - e_v)(1 - e_f)^2 \cdot (1 - P)$ |
| 2 | ✓ | ✗ | ✓ | $(1 - e_v)e_f(1 - e_f) \cdot 0.5$ |
| 3 | ✓ | ✓ | ✗ | $(1 - e_v)(1 - e_f)e_f \cdot 0.5$ |
| 4 | ✓ | ✗ | ✗ | $(1 - e_v)(e_f)^2 \cdot P$ |
| 5 | ✗ | ✓ | ✓ | $e_v(1 - e_f)^2 \cdot P$ |
| 6 | ✗ | ✗ | ✓ | $e_v e_f(1 - e_f) \cdot 0.5$ |
| 7 | ✗ | ✓ | ✗ | $e_v(1 - e_f)e_f \cdot 0.5$ |
| 8 | ✗ | ✗ | ✗ | $e_v(e_f)^2 \cdot (1 - P)$ |

Table 9: the possible error types with probabilities.

of both probe and galleries are correctly classified. In this case, our strategy gives us an error of $(1 - P)$. For type 2, the gender of probe and one of the gallery items is correctly classified, while the other gallery item is misclassified, we have an error of 0.5. If we go through all the cases, the total error $Prob(error)$ can be computed as

$$Prob(error) = 0.25 + 0.5 \sum_{type=1}^{8} Prob(error, type)$$
$$= 0.25 + 0.5(2e_f e_v - e_v - e_f + 1$$
$$+ P(2e_f + 2e_v - 4e_f e_v - 1))$$

Our objective is to minimize $Prob(error)$, so we must choose $P$ to minimize the above term. *I.e.* we must solve

$$\arg\min_{P} \ 2e_f e_v - e_v - e_f + 1 + P(2e_f + 2e_v - 4e_f e_v - 1)$$
$$\text{s.t. } 0 \leq P \leq 1.$$

Its easy to see that the solution for $P$ is 1.0 if its multiplicative factor is negative in the above equation, and 0 otherwise, *i.e.*

$$P = \begin{cases} 1, & \text{if } e_f + e_v < 2e_f e_v + 0.5 \\ 0, & \text{else} \end{cases}$$

The corresponding match error rates are $Prob(error) = 0.25 + 0.5(e_f + e_v - 2e_v e_f)$ and $0.75 + e_f e_f - 0.5(e_v + e_f)$ respectively.

Although complicated looking, the solution is, in fact, quite intuitive. When gender classification is either better than random for both modalities (*i.e.* $e_f, e_v > 0.5$) or worse than random for both ($e_f, e_v < 0.5$), the best strategy is to select the gallery item that matches the gender of the probe. If either of these is random (*i.e.* either $e_f$ or $e_v$ is 0.5) , the choice of $P$ does not matter, and the error is 0.5. If one of the two is correct more than half the time, and the other is wrong more than half the time (*e.g.* $e_f < 0.5$, $e_v > 0.5$), the optimal choice is to select the gallery item that is classified as *mismatched* in gender with the probe.

## B.2 Accuracy of 1:N matching based on gender

We now consider the best achievable performance on 1:$N$ matching, when the only information known is the gender of the voices and faces.

### B.2.1 Perfect gender identification

Consider the situation where the gender of the faces and voices in each test trial is perfectly known.

We employ the following strategy: we randomly select one of the gallery instances that have the same gender as the probe instance. If there are $K$ *imposter* gallery instances of the same gender as the probe instance, the expected *accuracy* is $\frac{1}{K+1}$. The probability of randomly having $K$ of $N-1$ imposters of the same gender as the probe is given by

$$Prob(K; N-1) = \binom{N-1}{K} 0.5^{N-1}$$

The overall accuracy is given by:

$$Prob(correct) = \sum_{K=0}^{N-1} \frac{Prob(K; N-1)}{K+1}$$
$$= 0.5^{N-1} \sum_{K=0}^{N-1} \binom{N-1}{K} \frac{1}{K+1}$$
$$= \frac{0.5^{N-1}}{N} \sum_{k=1}^{N} \binom{N}{K}$$
$$= \frac{0.5^{N-1}(2^N - 1)}{N}$$
$$= \frac{(2 - 0.5^{N-1})}{N},$$

giving us the error

$$P(error) = 1 - \frac{(2 - 0.5^{N-1})}{N}.$$

### B.2.2 IMPERFECT GENDER IDENTIFICATION

Consider now that the gender recognition is erroneous for voices with probability $e_v$ and for faces with probability $e_f$. Note that regardless of the error in gender recognition, the probability of any noisy gallery entry having any gender remains $0.5$.

To account for the possible error in gender classification, we consider the following stochastic policy: with probability $P$ we select one of the gallery entries with the same gender assigned to probe (by the gender classifier), and with probability $1 - P$ we choose one of the entries with the opposite gender assigned to the probe.

Let $\alpha$ represent the probability that the genders assigned to probe and the corresponding gallery entry by their respective classifiers are identical.

$$\alpha = e_v e_f + (1 - e_v)(1 - e_f).$$

The equation above considers both possibilities: that both the probe and its matching gallery entry are correctly classified, and that both of them are misclassified. It follows that the probability and its matching gallery entries are assigned different genders is $1 - \alpha$.

Given that we have selected the correct gender for retrieval from the the gallery (*i.e.* that the gender we have selected is the same as that assigned to the gallery entry matching the probe by the face classifier), using the same analysis as in Section B.2.1, we obtain the following probability of being correct:

$$P(correct|correct\ gender) = \frac{(2 - 0.5^{N-1})}{N}$$

The probability of selecting the correct gender is given by

$$P(correct\ gender) = P\alpha + (1 - P)(1 - \alpha)$$

Since the probability of being correct when we choose the wrong gender is 0, the overall probability of being correct is

$$P(correct) = (P\alpha + (1 - P)(1 - \alpha))\frac{(2 - 0.5^{N-1})}{N}$$

$$= (P(2\alpha - 1) + 1 - \alpha)\frac{(2 - 0.5^{N-1})}{N} \tag{4}$$

Maximizing the probability requires us to solve

$$\arg\max_P\ P(2\alpha - 1) + 1 - \alpha,\ \ \text{s.t.}\ \ 1 \geq P \geq 0$$

which gives us the optimal $P$ as

$$P = \begin{cases} 1 & \text{if}\ \alpha > 0.5 \\ 0 & \text{otherwise} \end{cases}$$

and the optimal error as

$$P(error) = \begin{cases} 1 - \alpha\frac{(2 - 0.5^{N-1})}{N}, & \text{if}\ \alpha > 0.5 \\ 1 - (1 - \alpha)\frac{(2 - 0.5^{N-1})}{N} & \text{otherwise.} \end{cases}$$

### B.3 EER OF VERIFICATION BASED ON GENDER

Here we show that the equal-error-rate for verification (determining if the the subjects in two recordings are the same) can be as high as 33%, through gender matching alone.

The problem is as follows: we are given a pair of inputs, one (features extracted from) a face, and the other a voice. We must determine whether they are both from the same speaker.

The test set include some number of "positives", where both do belong to the same subject, and some "negatives", where both do not. If a positive is falsely detected as a negative, we have an

instance of false rejection. If a negative is wrongly detected as a positive, we have an instance of false acceptance.

Let $F_R$ represent the 'false rejection rate", *i.e.* the fraction of all positives that are wrongly rejected. Let $F_A$ represent the "false acceptance rate", *i.e.* the fraction of negatives that are wrongly accepted. Any classifier can generally be optimized to trade off $F_R$ against $F_A$. The "Equal Error Rate" (EER) is achieved when $F_R = F_A$.

Among the "positive" test pairs, both voice and face in each pair have the same gender. We assume the "negative" test instances are drawn randomly, *i.e.*, 0.5 of all negative pairs have the same gender, while the remaining 0.5 do not.

### B.3.1   PERFECT GENDER IDENTIFICATION

Consider the situation where we know the subject's gender for both the voices and faces (or, alternately, are able to identify the gender from the voice or face perfectly).

We employ the following strategy: if the gender of the voice and face are different, we declare it as a negative 100% of the time. If the two are from the same gender, we randomly call it a positive $P$ of the time, where $0 \leq P \leq 1.0$.

Using this strategy, the false acceptance rate is:

$$F_A = 0.5 \times 0 + 0.5 \times P = 0.5P.$$

Here we're considering that using our strategy we never make a mistake on the 50% of negative pairs that have mismatched genders, but are wrong $P$ of the time on the negative pairs with matched genders.

Among the positives, where all pairs are gender matched, our strategy of accepting only a fraction $P$ of them as positives will give us a false rejection rate $F_R = 1 - P$.

The equal error rate is achieved when $F_R = F_A$, *i.e.*

$$0.5P = 1 - P,$$

giving us $P = \frac{2}{3}$, *i.e.* the best EER is achieved when we accept gender-matched pairs two-thirds of the time.

The EER itself is $0.5P = \frac{1}{3}$.

Thus, merely by being able to identify the gender of the subject accurately, we are able to verification EER of 0.33.

### B.3.2   IMPERFECT GENDER IDENTIFICATION

Now let us consider the situation where gender identification itself is imperfect, and we have error rates $e_f$ and $e_v$ in identifying the gender of the face and the voice, respectively. Assume these error rates are known.

To account for this, we modify our strategy: when we find the genders of the voice and face to match, we accept the pair as positive $P$ of the time, but when they are *mismatched* we still accept them as positive $Q$ of the time.

Let $\alpha$ represent the probability that we will correctly call the polarity of the gender match between the voice and the face. *I.e.* $\alpha$ is the probability that if the two have the same gender, we will correctly state that they have the same gender, or if they are of opposite gender, we will correctly state they are of opposite gender.

$$\alpha = (1 - e_f)(1 - e_v) + e_f e_v.$$

This combines two terms: that we call the genders of both the voice and face correctly, and that we call them both wrongly (which also results in finding the right polarity of the relationship). Its easy to see that $0 \leq \alpha \leq 1$, and to verify that when gender identification is perfect, $\alpha = 1.0$. The probability of calling the polarity of the gender relationship wrongly is $1 - \alpha$.

Among the positive test pairs, all pairs are gender matched. We will correctly call $\alpha$ of these as gender matched. Using our strategy, our error on these instances is $(1 - P)$. We will incorrectly call

$1 - \alpha$ of these as gender mismatched, and the error on these instances is $(1 - Q)$. So the overall false rejection rate is given by

$$F_R = \alpha(1 - P) + (1 - \alpha)(1 - Q) = 1 - \alpha P - (1 - \alpha)Q$$

Among the negative pairs, half are gender matched, and half are gender mismatched. Using the same logic as above, the error on the gender-matched negative pairs is $\alpha P + (1 - \alpha)Q$. Among the gender mismatched pairs the error is $\alpha Q + (1 - \alpha)P$. The overall false acceptance rate is given by

$$F_A = 0.5(\alpha P + (1 - \alpha)Q) + 0.5(\alpha Q + (1 - \alpha)P) = 0.5(P + Q).$$

Equating $F_A$ and $F_R$ as the condition for EER, we obtain

$$1 - \alpha P - (1 - \alpha)Q = 0.5(P + Q)$$
$$\implies (3 - 2\alpha)Q + (1 + 2\alpha)P = 2.$$

Since at EER, the EER equals $F_A$, and we would like to minimize it, we obtain the following solution to determine the optimal $P$ and $Q$:

$$\arg\min_{P,Q} P + Q$$
$$s.t. 1 \geq P, Q \geq 0, \quad (3 - 2\alpha)Q + (1 + 2\alpha)P = 2.$$

For $\alpha > 0.5$ it is easy to see that the solution to this is obtained at

$$Q = 0$$
$$P = \frac{2}{1 + 2\alpha}.$$

For $\alpha < 0.5$ the optimal solution is at

$$P = 0$$
$$Q = \frac{2}{3 - 2\alpha}.$$

That is, when the probe and gallery classifiers are likely to make the same error more than half the time, the optimal solution is to always reject pairs detected as having mismatched genders, and to accept matched-gender pairs $\frac{2}{1+2\alpha}$ of the time. The optimal EER is $\frac{1}{1+2\alpha}$.

When they are more likely to make different errors, the optimal solution is to always reject pairs detected as having matched genders, and to accept mismatched-gender pairs $\frac{2}{3-2\alpha}$ of the time. The optimal EER now is $\frac{1}{3-2\alpha}$.

Note that if an operating point other than EER were chosen to quantify performance (e.g. $F_A = \beta F_R$ for $\beta \neq 1$, or for some fixed $F_A$ or $F_R$), the above analysis can be modified to accommodate it, provided a feasible solution exists.

