# OpenReview forum: "Disjoint Mapping Network for Cross-modal Matching of Voices and Faces"
_ICLR.cc/2019/Conference_

### Official Review · AnonReviewer3 · 2018-11-03
**Networks that predict covariates of multimodal inputs like identity and gender produce better representations for cross-modal matching and retrieval tasks than directly predicting cross-modal matches.  Paper and well written and experiments are thorough.**

**Rating:** 7
**Confidence:** 4

**Review:**

This paper aims at matching people's voices to the images of their faces. It describes a method to train shared embeddings of voices and face images. The speech and image features go through separate neural networks until a shared embedding layer. Then a classification network is built on top of the embeddings from both networks.  The classification network predicts various combinations of covariates of faces and voices: gender, nationality, and identity.  The input to the classification network is then used as a shared representation for performing retrieval and matching tasks.

Compared with similar work from Nagrani et al (2018) who generate paired inputs of voices and faces and train a network to classify if the pair is matched or not, the proposed method doesn't require paired inputs.  It does, however, require inputs that are labeled with the same covariates across modalities.  My feeling is that paired positive examples are easier to obtain (e.g., from unlabeled video) than inputs labeled with these covariates, although paired negative examples require labeling and so may be as difficult to obtain.

Several different evaluations are performed, comparing networks that were trained to predict all subsets of identity, gender, and nationality.  These include identifying a matching face in a set of faces (1,2 or N faces) for a given voice, or vice versa. Results show that the network that predicts identity+gender tends to work best under a variety of careful examinations of various stratifications of the data.  These stratifications also show that while gender is useful overall, it is not when the gender of imposters is the same as that of the target individual.  The results also show that even when evaluating the voices and faces not shown in the training data, the model can achieve 83.2% AUC on unseen/unheard individuals, which outperforms the state-of-the-art method from Nagrani et al (2018).

An interesting avenue of future work would be using the prediction of these covariates to initialize a network and then refine it using some sort of ranking loss like the triplet loss, contrastive loss, etc.


Writing:
* Overall, ciations are all given in textual form Nagrani et al (2018) (in latex this is \citet{} or \cite{}), when many times parenthetical citations (Nagrani et al, 2018) (in latex this is \citep{}) would be more appropriate.
* The image of the voice waveform in Figures 1 and 2 should be replaced by log Mel-spectrograms in order to illustrate the network's input.
* "state or art" instead of "state-of-the-art" on page 3.
* In subsection 2.4: "mGiven" is written instead of "Given".
* On Page 6 Section 3.1 "1:2 matching" paragraph. "Nagrani et al." is written twice. * * Page 6 mentions that there is a row labelled "SVHF-Net" in table 2, but there is no such row is this table.
* Page 7 line 1, “G,N” should be "G, N".

---

> ### Author Response · Authors · 2018-11-16
> **Rebuttal for reviewer 3**
>
> We thank the reviewer for the very positive and encouraging review.
>
> Q1. My feeling is that paired positive examples are easier to obtain (e.g., from unlabeled video) than inputs labeled with these covariates, although paired negative examples require labeling and so may be as difficult to obtain.
>
> A1. We agree with the reviewer. Compared to covariates, the pairwise label is usually easier to obtain. However, some challenges still exist for collecting the examples from video, making it a non-trivial problem. For example, the cases of reaction shots, flashbacks and dubbing in videos may result in noisy labels. Previous work [6] investigated the use of the paired data in self-supervised learning manner, where SyncNet [7] is adopted to obtain the speaking faces.
>
> For our paper, we focus on proposing a DIMNet framework to learn embeddings for cross-modal matching with the given cross-modal data and their labeled covariates. How to collect data is perhaps beyond the scope of this paper but could be an interesting direction for our future work.
>
> Q2. Typos
> A2. We thank the reviewer for pointing out the typos. All the typos are fixed in the updated paper.
> Citations: we have carefully checked the citations and accordingly fixed them one by one .
> Figures: The waveforms have been replaced by log Mel-spectrograms.
> “state or art” -> “state-of-the-art”
> “mGiven” -> “Given”
> "Nagrani et al. Nagrani et al. (2018b)" -> “Nagrani et al. (2018b)”;  typo in Table 2 is fixed
> “G,N” -> "G, N"
>
> [6] Nagrani, Arsha, Samuel Albanie, and Andrew Zisserman. "Learnable PINs: Cross-Modal Embeddings for Person Identity." arXiv preprint arXiv:1805.00833 (2018).
> [7] Chung, Joon Son, and Andrew Zisserman. "Out of time: automated lip sync in the wild." Asian Conference on Computer Vision. Springer, Cham, 2016.

---

### Official Review · AnonReviewer2 · 2018-11-03
**Review of Disjoint Mapping Network for Cross-modal Matching of Voices and Faces**

**Rating:** 6
**Confidence:** 3

**Review:**

# Summary

The article proposes a deep learning-based approach aimed at matching face images to voice recordings belonging to the same person.

To this end, the authors use independently parametrized neural networks to map face images and audio recordings -- represented as spectrograms -- to embeddings of fixed and equal dimensionality. Key to the proposed approach, unlike related prior work, these modules are not directly trained on some particular form of the cross-modal matching task. Instead, the resulting embeddings are fed to a modality-agnostic, multiclass logistic regression classifier that aims to predict simple covariates such as gender, nationality or identity. The whole system is trained jointly to maximise the performance of these classifiers. Given that (face image, voice recording) pairs belonging to the same person must share equal for these covariates, the neural networks embedding face images and audio recordings are thus indirectly encouraged to map face images and voice recordings belonging to the same person to similar embeddings.

The article concludes with an exhaustive set of experiments using the VGGFace and VoxCeleb datasets that demonstrates improvements over prior work on the same set of tasks.

# Originality and significance

The article follows-up on recent work [1, 2], building on their original application, experimental setup and model architecture. The key innovation of the article, compared to the aforementioned papers, lies on the idea of learning face/voice embeddings to maximise their ability to predict covariates, rather than by explicitly trying to optimise an objective related to cross-modal matching. While the fact that these covariates are strongly associated to face images and audio recordings had already been discussed in [1, 2], the idea of actually using them to drive the learning process is novel in this particular task.

While the article does not present substantial, general-purpose methodological innovations in machine learning, I believe it constitutes a solid application of existing techniques. Empirically, the proposed covariate-driven architecture is demonstrated to lead to better performance in the (VGGFace, VoxCeleb) dataset in a comprehensive set of experiments. As a result, I believe the article might be of interest to practitioners interested in solving related cross-modal matching tasks.

# Clarity

The descriptions of the approach, related work and the different experiments carried out are written clearly and precisely. Overall, the paper is rather easy to read and is presented using a logical, easy-to-follow structure.

In my opinion, perhaps the only exception to that claim lies in Section 3.4. If possible, I believe the Seen-Heard and Unseen-Unheard scenarios should be introduced in order to make the article self-contained.

# Quality

The experimental section is rather exhaustive. Despite essentially consisting of a single dataset, it builds on [1, 2] and presents a solid study that rigorously accounts for many factors, such as potential confounding due to gender and/or nationality driving prediction performance in the test set.

Multiple variations of the cross-modal matching task are studied. While, in absolute terms, no approach seems to have satisfactory performance yet, the experimental results seem to indicate that the proposed approach outperforms prior work.

Given that the authors claimed to have run 5 repetitions of the experiment, I believe reporting some form of uncertainty estimates around the reported performance values would strengthen the results.

However, I believe that the success of the experimental results, more precisely, of the variants trained to predict the "covariate" identity, call into question the very premise of the article. Unlike gender or nationality, I believe that identity is not a "covariate" per se. In fact, as argued in Section 3.1, the prediction task for this covariate is not well-defined, as the set of identities in the training, validation and test sets are disjoint. In my opinion, this calls into question the hypothesis that what drives the improved performance is the fact that these models are trained to predict the covariates. Rather, I wonder if the advantages are instead a "fortunate" byproduct of the more efficient usage of the data during the training process, thanks to not requiring (face image, audio recording) pairs as input.

# Typos

Section 2.4
1) "... image.mGiven ..."
2) Cosine similarity written using absolute value |f| rather than L2-norm ||f||_{2}
3) "Here we are give a probe input ..."

# References

[1] Nagrani, Arsha, Samuel Albanie, and Andrew Zisserman. "Learnable PINs: Cross-Modal Embeddings for Person Identity." arXiv preprint arXiv:1805.00833 (2018).
[2] Nagrani, Arsha, Samuel Albanie, and Andrew Zisserman. "Seeing voices and hearing faces: Cross-modal biometric matching." Proceedings of the IEEE Conference on Computer Vision and Pattern Recognition. 2018.

---

> ### Author Response · Authors · 2018-11-16
> **Rebuttal for reviewer 2**
>
> We sincerely appreciate the review for the recognition of our novelty and many valuable suggestions.
>
> Our main contribution mainly lies in proposing a cross modal matching framework called DIMNet, which learns a shared representation for different modalities by mapping them individually to their common covariates. Our basic intuition is that if the learned embeddings of voices and faces can be correctly classified by a unified (linear) classifier, the embeddings of the same class should be in a common decision region and close to each other.
> Compared to the existing work [3,4], the supervision could be any combination of covariates, which enables us to isolate and analyze the effect of the individual covariate to the learned embeddings. Moreover, DIMNet makes better use of the multiple covariates in the course of training.
>
> In order to perform fair comparisons, we exactly follow the experimental setup in pioneering work [3,4], and achieve significant improvements compared to these strong baselines [3,4].
>
> Q1. In my opinion, perhaps the only exception ... in order to make the article self-contained.
> A1. We thank the reviewer for this suggestion. We do mention the two scenarios in the paper, but the reviewer is right, we do not explicitly introduce them. We now do so in the updated paper.
>
> In summary, the audio data we used in Section 3.4 is the same as those in other experiment sections, while the visual data is extracted from the video frames in VoxCeleb dataset at 25/6 fps. For fair comparison, we follow the train/val/test split strategy from [4] and evaluate our DIMNet models under Seen-Heard (closed-set) and Unseen-Unheard (open-set)scenarios. More details can be found in the updated paper.
>
> Action taken: Provided more details about the datasets, and experimental settings in Section 3.4 and appendix A.
>
>
> Q2. Given that the authors claimed to have run 5 repetitions ... strengthen the results.
> A2. We thank the reviewer for this suggestion. We have now computed the standard deviations of the results and added them to each table.
>
> Action taken: Added standard deviations of the results to each table.
>
> Q3. However, I believe that the success of the experimental results, ..., validation and test sets are disjoint.
> A3. Our definition of covariate, as stated in the paper, are the ID-sensitive factors that can simultaneously affect voice and face, e.g. nationality, gender, identity, etc. We do not require the value these factors take to be the same between the training and test set. Thus, from the perspective of our model, we only require that faces and voices in the test set co-vary with ID; we do not require that ID to be present in training. What we are learning is the nature of the covariation with the variable in general, not merely the covariation with the specific values the variable takes in the training set.
>
> To give another example, if we were to consider age as a covariate (which we have not in the current set of experiments, since we do not desire age-sensitive matching), we would expect to learn how both voice and face embeddings vary with age. This then could be used to match voice and face embeddings in the test set even if the corresponding age were not observed in training.
>
> Action taken: Added the above discussions about covariates to introduction section.
>
> Q4. In my opinion, this calls into question the hypothesis ..., thanks to not requiring (face image, audio recording) pairs as input.
> A4. More efficient usage of the data is indeed one of the advantages of our DIMNet framework, as we state in both the introduction and the discussion. And this is achieved, by design, by exploiting (and explicitly modelling) the dependence between the modalities and covariates in a generalizable manner. The outcomes we observe in our experiments are entirely to be expected, from our hypothesis, and we believe that the rather detailed set of experiments (and the analyses in our appendix) show that the results are not merely fortuitous. As indicated by our experiments, DIMNet-I achieves 83.45% accuracy on 1:2 matching task since ID is undoubtedly the most informative covariate. Even using less informative covariates, DIMNet-G still achieves 72% matching accuracy.
>
> Q5. Typos
> A5. We thank the reviewer for the pointing out the typos. All the typos are fixed in the updated paper.
> "... image.mGiven ..." -> "... image. Given ..."
> |Fv||Ff| -> ||Fv||_2||Ff||_2
> "Here we are give a probe input ..." -> Here we are given a probe input …”
>
> [3] Nagrani, Arsha, et al. "Seeing voices and hearing faces: Cross-modal biometric matching." IEEE CVPR 2018.
> [4] Nagrani, Arsha, et al. "Learnable PINs: Cross-Modal Embeddings for Person Identity." arXiv preprint arXiv:1805.00833 (2018).
> [5] Chung, Joon Son, et al. "Out of time: automated lip sync in the wild." ACCV, 2016.

---

### Official Review · AnonReviewer1 · 2018-11-04
**Covariates factors are learned from voice and image data using CNNs. A logistic classifier is trained for cross-modal matching from covariates.**

**Rating:** 7
**Confidence:** 4

**Review:**

Authors aim to reveal relevant dependencies between voice and image data (under a cross-modal matching framework) through common covariates (gender, ID, nationality). Each covariate is learned using a CNN from each provided domain (speak recordings and face images), then, a classifier is determined from a shared representation, which includes the CNN outputs from voice-based and image-based covariate estimations. The idea is interesting, and the paper ideas are clear to follow.

Pros:
- New insights to support cross-modality matching from covariates.
- Competitive results against state-of-the-art.
-Convincing experiments.

Cons:
-Fixing the output dimension to d (for both voice and image-based CNN outputs) could lead to unstable results. Indeed, the comparison of voice and face-based covariate estimates are not entirely fair due to the intrinsic dimensionality can vary for each domain. Alternatives as canonical correlation analysis can be coupled to joint properly both domains.
- Table 4 - column ID results are not convincing (maybe are not clear for me).

---

> ### Author Response · Authors · 2018-11-16
> **Rebuttal for reviewer 1**
>
> We thank the reviewer for the recognition of the novelty and the detailed experimental evaluation in our contribution.
>
> Q1. Fixing the output dimension to d (for both voice and image-based CNN outputs) could lead to unstable results. Indeed, the comparison of voice and face-based covariate estimates are not entirely fair due to the intrinsic dimensionality can vary for each domain. Alternatives as canonical correlation analysis can be coupled to joint properly both domains.
> A1. In order to compare embeddings from two modalities (domains), the dimensionality of the embeddings need to be the same. We agree with the reviewer that the intrinsic dimensionality of data in different modalities (domains) could vary. However, it does not contradict the fact that these data can be well represented by the identical-dimensioned embeddings through CNNs, and most importantly, the performance (in the following table) is very stable within a wide range of embedding dimension, showing that the accuracy is not sensitive to the embedding dimension. The idea of using the identical-dimensioned embeddings is also adopted by [1] and [2].
>
> The accuracies of DIMNet-I with different embedding dimensions on 1:2 matching experiments
> -------------------------------------------------------------------------------
> Dimension       32           64          128         256         512
> -------------------------------------------------------------------------------
> DIMNet-I       82.20      83.45      83.87      83.43      83.16
> -------------------------------------------------------------------------------
>
> Action taken: Added this experiment in appendix A with analysis.
>
> Canonical correlation analysis (CCA) is a good idea to investigate the correlation of data between different domains, and it could indeed be used to match different-dimensioned embeddings derived from the two modalities, and was indeed one of our ideas enroute to the development of DIMNet. The reasons we do not use it are the following: (a) The final projection in CCA is a linear transform that is easily subsumed within the network (in fact a linear projection may be viewed as a fully-connected layer with linear activations).  (b) More importantly, the underlying idea of CCA is very different from DIMNet. Specifically, CCA requires one-to-one correspondence between the two modalities it considers, an assumption DIMNet explicitly tries to avoid. Specifically, in the case of static face images vs. voice samples, it is unclear that such correspondence is derivable. Given that we have multiple face images and multiple voice recordings for any person, all captured at different times, which pairs of voice recordings and face images would we group together? Any correspondence imposed would be artificial. On the other hand, DIMNet builds correspondences between voices (or faces) and their covariates, and does not expect direct correspondence between the two modalities -- in fact this is one of the key features of our model which differentiates it from prior work. The comparison could be more intuitively noted from Fig. 1 in our paper.
>
> Q2. Table 4 - column ID results are not convincing (maybe are not clear for me).
> A2. The ID column in Table 4 shows the mean average precision (mAP) of the retrieved ID, when one modality (e.g. face) is posed as the query and retrieval of corresponding recordings of the other modality (e.g. voice) must be performed. The evaluation dataset consists of 21,799 voices and 58,420 faces, both from 182 identities. Compared to gender (2 classes) and nationality (unbalanced 28 classes), it is a challenging problem to rank the gallery voices (faces) based on the probe face (voice) given these many identities (182 classes). Chance-level performance (i.e., random guess) is about 0.55% for voice->face and 0.58% for face->voice, while we achieved 1.07~4.25% for voice->face and 1.03%~4.17% for face->voice. It means that the DIMNet models do learn useful associations between voices and faces.
>
> Action taken: Added one row of chance level results to Table 4 with analysis.
>
> [1] Nagrani, Arsha, Samuel Albanie, and Andrew Zisserman. "Learnable PINs: Cross-Modal Embeddings for Person Identity." arXiv preprint arXiv:1805.00833 (2018).
> [2] Kim, Changil, et al. "On Learning Associations of Faces and Voices." arXiv preprint arXiv:1805.05553 (2018).

---

### Meta-Review · Area_Chair1 · 2018-12-20
**metareviw**

**Confidence:** 5
**Recommendation:** Accept (Poster)

**Metareview:**

All reviewers agree that the proposed method interesting and well presented. The authors' rebuttal addressed all outstanding raised issues. Two reviewers recommend clear accept and the third recommends borderline accept. I agree with this recommendation and believe that the paper will be of interest to the audience attending ICLR. I recommend accepting this work for a poster presentation at ICLR.